# Unified Multimodal Visual Tracking with Dual Mixture-of-Experts

Lingyi Hong [1]   Jinglun Li [2]   Xinyu Zhou [3]   Kaixun Jiang [4]   Pinxue Guo [4]   Zhaoyu Chen [4]   Runze Li [5]
Xingdong Sheng [5]   Wenqiang Zhang [1 4]

## Abstract

Multimodal visual object tracking can be divided into to several kinds of tasks (e.g. RGB and RGB+X tracking), based on the input modality. Existing methods often train separate models for each modality or rely on pretrained models to adapt to new modalities, which limits efficiency, scalability, and usability. Thus, we introduce *OneTrackerV2*, a unified multi-modal tracking framework that enables end-to-end training for any modality. We propose Meta Merger to embed multi-modal information into a unified space, allowing flexible modality fusion and robustness. We further introduce Dual Mixture-of-Experts (DMoE): T-MoE models spatio-temporal relations for tracking, while M-MoE embeds multi-modal knowledge, disentangling cross-modal dependencies and reducing feature conflicts. With a shared architecture, unified parameters, and a single end-to-end training, *OneTrackerV2* achieves state-of-the-art performance across five RGB and RGB+X tracking tasks and 12 benchmarks, while maintaining high inference efficiency. Notably, even after model compression, *OneTrackerV2* retains strong performance. Moreover, *OneTrackerV2* demonstrates remarkable robustness under modality-missing scenarios.

## 1. Introduction

Visual object tracking (Zhou et al., 2023a;b; Bertinetto et al., 2016; Chen et al., 2020; Li et al., 2019a; Wu et al., 2013; Ye

[1]Shanghai Key Lab of Intelligent Information Processing, College of Computer Science and Artificial Intelligence, Fudan University [2]JIIOV Technology [3]School of Mechanical and Aerospace Engineering, Nanyang Technological University [4]College of Intelligent Robotics and Advanced Manufacturing, Fudan University [5]Lenovo Research. Correspondence to: Lingyi Hong <honglyhly@gmail.com>, Wenqiang Zhang <wqzhang@fudan.edu.cn>.

*Proceedings of the 43ʳᵈ International Conference on Machine Learning*, Seoul, South Korea. PMLR 306, 2026. Copyright 2026 by the author(s).

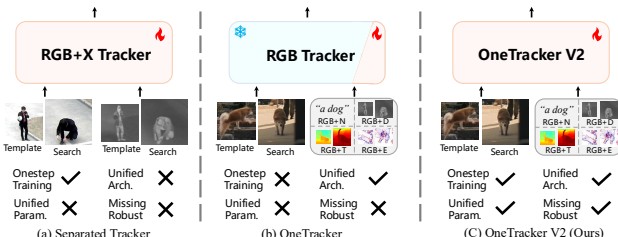

*Figure 1.* **Comparison of our OneTracker V2 and previous models.** (a) **Separated trackers**: task-specific architectures trained independently for each task. (b) **Fine-tuned trackers**: represented by OneTracker (Hong et al., 2024b), which adapts pretrained RGB trackers to downstream RGB+X tasks through fine-tuning. (c) *OneTrackerV2* **(Ours)**: a unified architecture with shared parameters, trained once to handle multiple multimodal tracking tasks.

et al., 2022; Chen et al., 2021) aims to localize target object in each subsequent frame of a video based on its appearance in the first frame. Depending on input modalities, visual object tracking can be categorized into several types (Hong et al., 2024b), such as RGB tracking (where only RGB images are available) (Chen et al., 2023b;a; Wei et al., 2023; Bai et al., 2024; Zhou et al., 2025a) and RGB+X tracking (where additional modalities are incorporated alongside RGB images), which can be further divided into RGB+D (Depth) tracking (Yan et al., 2021b; Kristan et al., 2022), RGB+T (Thermal) tracking (Li et al., 2021; 2019b), RGB+E (Event) tracking (Wang et al., 2023), and RGB+N (Language) tracking (Li et al., 2017; Fan et al., 2019).

As illustrated in Figure 1, existing multimodal trackers can be categorized into three categories: (a) separated trackers, which design and train task-specific architectures for each modality; (b) adaptation-based trackers such as OneTracker (Hong et al., 2024b), which fine-tune pretrained RGB trackers for downstream multimodal tasks; and (c) preliminary unified trackers like SUTrack (Chen et al., 2025), which attempt to handle multiple modalities within a single model. Despite their success, these approaches suffer from several limitations. (1) **Multi-step training**, where transferring pretrained models leads to suboptimal convergence(Zhu et al., 2023; Hong et al., 2024b; Hou et al., 2024); (2) **Lack of a unified architecture**, requiring handcrafted, task-specific designs; (3) **Un-unified Parameters**, where even shared architectures (Hong et al., 2024b) rely on task-dependent weights; (4) **Vulnerability to missing modali-**

**ties**, caused by heavy reliance on fixed input configurations, and (5) **Feature Conflict**, where heterogeneous information is entangled. Simply concatenating multimodal tokens forces the model to learn spatio-temporal motion and modality-specific patterns simultaneously within a shared space, leading to optimization interference and reduced discriminability (Chen et al., 2025).

To address these issues, we introduce a unified training paradigm that enables a single tracker to handle diverse modalities in a scalable and robust manner. Instead of designing separate architectures or relying on multi-stage fine-tuning, our approach emphasizes training from the ground up with a shared backbone, consistent parameterization, and modality-robust fusion. A key component of this framework is Meta Merger, which embeds all modalities into a joint representation space and facilitates flexible, learnable interactions between RGB and auxiliary modalities. Unlike prior methods that either double computation with separate branches or apply naive concatenation (Chen et al., 2025), our unified design achieves: (1) **Comprehensive cross-modal interaction**, where the meta embedding captures global context across modalities; (2) **Robustness to missing modalities**, by dynamically adapting to incomplete inputs; and (3) **Parameter efficiency**, as a single set of parameters that can generalize across all tasks.

After obtaining the integrated meta embeddings, we feed them into a vision transformer for relation modeling. To enhance its representational capacity and resolve the aforementioned feature conflicts, we introduce Dual Mixture-of-Experts (DMoE). Specifically, the T-MoE is dedicated to capturing complex and diverse spatio-temporal matching patterns, while the M-MoE focuses on multimodal knowledge fusion. Unlike generic MoE designs, we employ an orthogonality-driven decoupling loss and a router clustering loss to force these experts into distinct functional subspaces. This formulation provides two critical advantages: (1) **Explicit Feature Decoupling**: By separating temporal matching from multimodal fusion, the model avoids optimizing heterogeneous and often conflicting objectives within a shared parameter space, significantly reducing interference. (2) **Sparsely Activated Efficiency**: The DMoE structure allows for substantial capacity expansion to handle diverse tracking scenarios while maintaining near-constant inference computation, ensuring that OneTrackerV2 remains competitive in both accuracy and speed.

Extensive experiments validate the effectiveness of our *OneTrackerV2*. On RGB tracking benchmarks, previous RGB-only trackers can only perform RGB tracking; by contrast, *OneTrackerV2* can handle multimodal tracking and yet still outperforms those RGB-only baselines on RGB tracking tasks. For RGB+X tracking benchmarks, previous methods typically require task-specific architectures or a two-stage

fine-tuning pipeline, while *OneTrackerV2* surpasses these approaches, only requiring a single step training and a unified architecture with shared parameters. Further analyses on robustness and model compression demonstrate that *OneTrackerV2* maintains high accuracy under missing modalities and compressed settings, highlighting its scalability, robustness, and efficiency.

In summary, the contributions can be summarized as follows: (1) We introduce *OneTrackerV2*, a unified framework that supports diverse multimodal tracking tasks through one-step training with a shared architecture and parameters, enabling scalability, robustness, and generalization across modalities. (2) We propose Meta Merger module to aggregate RGB and multimodal features into a unified space, enabling efficient and effective cross-modal interaction while maintaining robustness under missing-modality conditions. (3) We propose DMoE, which decouples spatio-temporal relation modeling from multimodal feature embedding, which expands representational space and improves performance with negligible computational overhead. (4) Extensive experiments demonstrate that *OneTrackerV2* achieves state-of-the-art performance across 5 tracking tasks and 12 benchmarks. Moreover, *OneTrackerV2* maintains high efficiency and exhibits strong robustness under model compression and modality-missing scenarios.

## 2. Related Works

**RGB Tracking.** Visual object tracking aims to localize the target object in each video frame based on its initial appearance. In RGB tracking, only RGB images are used as input. Early approaches (Bertinetto et al., 2016; Bolme et al., 2010; Chen et al., 2021; Danelljan et al., 2019; Henriques et al., 2014; Yan et al., 2021a; Zhang et al., 2020) adopted a two-stream pipeline, where feature extraction and relation modeling were performed separately. Recently, one-stream pipelines (Bai et al., 2024; Chen et al., 2022; 2023b; Cui et al., 2022; 2023; Gao et al., 2023; Ye et al., 2022; Zhou et al., 2023a; 2024; Zheng et al., 2024; Liang et al., 2025; Hong et al., 2024b; Zhou et al., 2025b; Chen et al., 2023a) have become dominant, integrating these two stages into a unified process. These models are built upon Vision Transformers with stacked encoder layers. However, these approaches remain restricted to RGB-only inputs. In contrast, our *OneTrackerV2* can process diverse multimodal inputs without relying on task-specific architectures.

**RGB+X Tracking.** In certain scenarios, relying solely on RGB images can be limiting. Thus, RGB+X tracking tasks are introduced, where additional modalities are incorporated to complement the weaknesses of RGB-based tracking. Depending on the input modality, RGB+X tracking can be categorized into four types: RGB+D (Depth) (Yan et al., 2021b; Kristan et al., 2022), RGB+T (Thermal) (Li et al.,

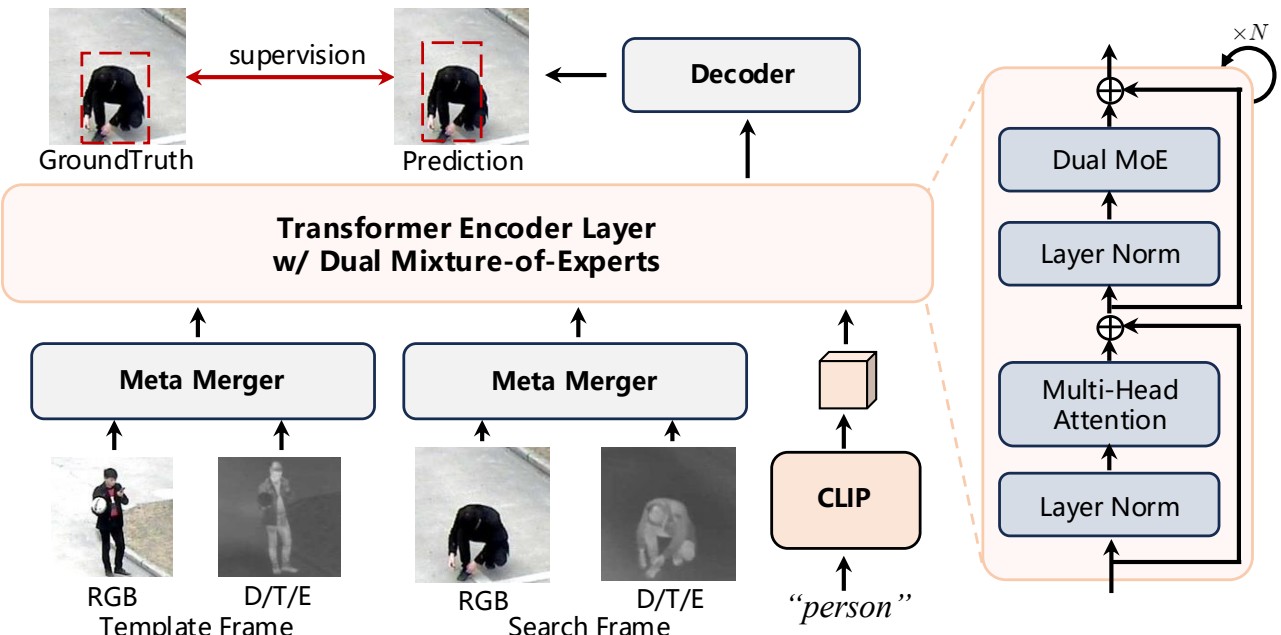

*Figure 2.* **Architecture of *OneTrackerV2*.** *OneTrackerV2* employs a Meta Merger to project diverse modalities into a shared embedding space. DMoE models spatio-temporal relations while decoupling multimodal dependencies.

2021; 2019b), RGB+E (Event) (Wang et al., 2023), and RGB+N (Language) tracking (Li et al., 2017; Fan et al., 2019; Wang et al., 2021).

Existing RGB+X tracking methods primarily rely on separated architectures, where task-specific networks (Yan et al., 2021b; Li et al., 2021; Wang et al., 2023; 2021; Su et al., 2026) are designed and trained independently for each modality combination. While effective, this strategy requires extensive architectural customization and separate training for every task. To alleviate this, subsequent works (Zhu et al., 2023; Hong et al., 2024b; Hou et al., 2024; Guo et al., 2024; Bai et al., 2024; Su et al., 2026) adapt pretrained RGB trackers to downstream RGB+X tasks through fine-tuning, enabling cross-modality transfer. However, these methods depend on multi-stage training and often suffer from sub-optimal performance. SUTrack (Chen et al., 2025) attempts to handle multiple modalities within a single model, but it remains vulnerable to performance degradation under missing-modality scenarios. Thus, we introduce ***OneTrackerV2***, a unified multimodal tracking framework that achieves robust performance without task-specific designs or multi-stage training process.

**Mixture of Experts.** Mixture-of-experts (MoE) increases model capacity via conditional computation over multiple specialized experts and has been widely applied in large language models (Dai et al., 2024; Fedus et al., 2022; Yang et al., 2025). In visual object tracking, several studies have explored MoE (Tan et al., 2025; Cai et al., 2025; Tan et al., 2024; Guo et al., 2024), typically either using MoE to trans-

fer pretrained RGB trackers to RGB+X settings or restricting training to RGB-only inputs. In contrast, we introduce Dual Mixture-of-Experts (DMoE) that combines a shared expert with a T-MoE and a M-MoE, explicitly decoupling temporal dynamics from multimodal integration. Equipped with DMoE, ***OneTrackerV2*** operates within a unified architecture with shared parameters and supports both RGB and RGB+X tracking with a one-step training.

## 3. OneTracker V2

### 3.1. Overall Architecture

As shown in Figure 2, ***OneTrackerV2*** is a unified framework that processes template and search regions from diverse modalities (e.g., RGB, Depth, Thermal) within a single architecture. To address the challenge of heterogeneous inputs, we first employ Meta Merger, which maps RGB and multimodal features into a shared embedding space via learnable meta embedding, ensuring a consistent representation before relation modeling. This unified representation is then fed into a Vision Transformer backbone integrated with our Dual Mixture-of-Experts (DMoE) module. DMoE employs two specialized branches—T-MoE for spatio-temporal matching and M-MoE for multimodal integration—to explicitly disentangle motion cues from modality-specific features. This decoupling effectively mitigates feature conflicts that typically arise in naive unified backbones. By utilizing shared parameters and a one-step training pipeline, ***OneTrackerV2*** achieves a scalable and robust solution for multimodal tracking without task-specific branches.

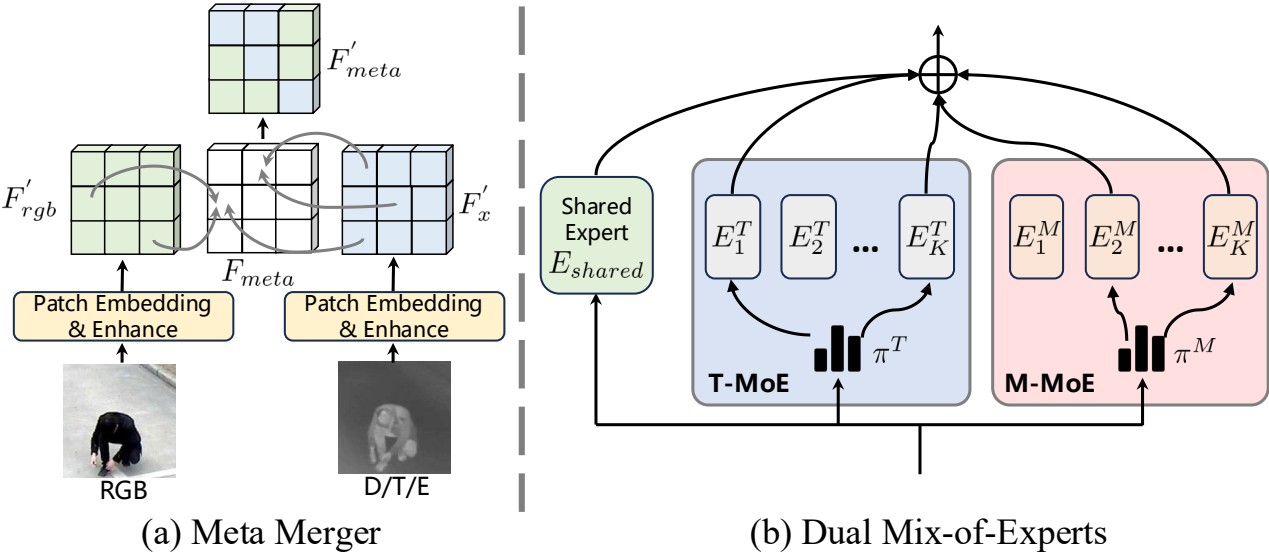

(a) Meta Merger                    (b) Dual Mix-of-Experts

*Figure 3.* (a) **Meta Merger.** Meta Merger unifies RGB and multimodal input into a shared space. (b) **Dual Mixture-of-Experts.** We introduce DMoE to decouple spatio-temporal relation modeling and multimodal feature integration.

## 3.2. Meta Merger

A central challenge in multimodal tracking is integrating heterogeneous modalities into a shared representation while maintaining robustness to missing data. Unlike task-specific branches (Hou et al., 2024) that increase complexity, or naive concatenation (Chen et al., 2025) that leads to feature entanglement, we introduce the Meta Merger(as shown in Figure 3(a)), a lightweight yet effective module to unify multimodal information into a shared embedding and support robust, modality-agnostic interaction.

Specifically, given the RGB frame and X modality frame, we first utilize a shared patch embedding layer and obtain corresponding feature maps $F_{rgb}$ and $F_x$. It is worth noting that for RGB tracking task, we use the same RGB images to replace the X images. Then, spatial and channel attentions are adopted to enhance both features, which can be illustrated as following:

$$F_{rgb}^{avg} = \text{AvgPool}(F_{rgb}), F_{rgb}^{max} = \text{MaxPool}(F_{rgb})$$
$$W_{rgb}^{spatial} = \sigma(\text{Conv}(F_{rgb}^{avg}) + \text{Conv}(F_{rgb}^{max})),$$
$$W_{rgb}^{channel} = \sigma(\text{Linear}(F_{rgb}^{avg}) + \text{Linear}(F_{rgb}^{max})),$$
$$F_{rgb}' = F_{rgb} \odot W_{rgb}^{spatial} \odot W_{rgb}^{channel} + F_{rgb},$$
$$(1)$$

where $\odot$ denotes element-wise multiplication, $W_{rgb}^{spatial}$ and $W_{rgb}^{channel}$ act as the spatial and channel attention weights respectively. For $F_x$, we also apply the same operations, which are omitted here. After that, a learnable meta embedding $F_{meta}$ is introduced as a global mediator to interact with both RGB and multimodal features:

$$F_{meta}' = \text{Conv}(\text{Conv}(F_{meta} + F_{rgb}')$$
$$+ \text{Conv}(F_{meta} + F_x') + F_{meta}). \quad (2)$$

Our Meta Merger leverages meta embedding as a central information hub. The meta embedding interacts with both RGB and auxiliary features through lightweight convolution transformations. This interaction allows the meta embedding to absorb, align, and redistribute cross-modal information, resulting in a compact and modality-agnostic semantic representation. Our Meta Merger features several advantages: (1) **Unified Modalities Bridge.** Meta embedding serves as a learnable bridge that harmonizes heterogeneous inputs, ensuring effective multimodal information alignment. (2) **Global Semantic Integration.** Meta embedding distills the most salient cues into a compact, globally consistent representation that benefits relation modeling downstream. (3) **Robustness to Missing Modalities.** Since fusion is centered on meta embedding rather than direct concatenation, Meta Merger remains stable and effective even when some modalities are absent, which is unavaible in previous works (Chen et al., 2025). (4) **Lightweight.** Compared to multi-branch designs (Hou et al., 2024) that double computational costs, our Meta Merger introduces minimal overhead while enabling scalable multimodal learning.

## 3.3. Dual Mixture-of-Experts

**Structure of DMoE.** After fusing multimodal features via Meta Merger, a Vision Transformer backbone is utilized for relation modeling. We introduce Dual Mixture-of-Experts (DMoE), which is shown in Figure 3 (b). Different from previous MoE designs (Fedus et al., 2022; Dai

et al., 2024), which typically focus on a single objective, our DMoE explicitly decouples spatio-temporal relation modeling and multimodal feature integration within the same framework. DMoE consists of three kinds of experts: shared expert $E_{shared}$, T-MoE (Temporal-MoE), and M-MoE (Multimodal-MoE). For each token $x \in \mathbb{R}^d$, DMoE computes its output as:

$$y = E_{shared}(x) + \underbrace{\sum_{i \in S_k^T} \hat{g}_i^T(x) \cdot E_i^T(x)}_{y^T} + \underbrace{\sum_{i \in S_k^M} \hat{g}_i^M(x) \cdot E_i^M(x)}_{y^M},$$
(3)

where $S_k^T = \text{top}-k(g_i^T(x))$ is the selected $k$ experts for T-MoE, $g_i^T(x) = \frac{\exp(\pi_i^T(x))}{\sum_{j=1}^{K} \exp(\pi_j^T(x))}$ denotes the gating weights after softmax, $\pi_i^T(\cdot)$ is origin gating value of T-MoE for the $i$-th expert, $\hat{g}_i^T(x) = \frac{g_i^T(x)}{\sum_{j \in S_k^T} g_j^T(x))}$ is the renormalized gating weights of $\text{top}-k$ experts, and $K$ is the number of experts. The gating process for M-MoE is defined analogously with $\pi_i^M(x)$, $g_i^M(x)$, and $\hat{g}_i^M(x)$. Each expert $E$ itself is implemented as a simple low-rank projection: inputs are mapped into a latent dimension $r$, transformed non-linearly, and then projected back. This design ensures high capacity without incurring prohibitive costs.

**Expert Decoupling.** If optimized jointly, T-MoE and M-MoE may collapse into learning overlapping patterns. To avoid redundancy, we introduce a dissimilarity loss to penalize cosine similarity between outputs of T-MoE ($y^T$) and M-MoE ($y^M$):

$$\mathcal{L}_{dis} = (\cos(y^T, y^M))^2 = (\frac{<y^T, y^M>}{||y^T||||y^M||})^2. \quad (4)$$

Intuitively, through $\mathcal{L}_{dis}$, we explicitly encourages T-MoE and M-MoE to remain decorrelated and enhance the diversity of learned features.

**Multimodal Router Cluster.** Although $\mathcal{L}_{dis}$ promotes diversity between T-MoE and M-MoE, the router of M-MoE remains disorganized across different modalities, failing to learn modality-specific expert assignments. To address this, we propose a router weights clustering regularization that aligns similar distributions to samples from the same modality and distinct distributions to samples from different modalities. Concretely, we compute similarity scores between routing logits $g^M(x)$ and encourage: (1) higher similarity for samples from the same modality; and (2) lower similarity for samples from different modalities.

Specifically, given the router logits $g^M(x_i) \in \mathbb{R}^K$ for a sample $x_i$ in a batch of size B, we compute the similarity matrix among routing distributions as $S_{ij} = <g^M(x_i), g^M(x_j)>$.

Then, we construct same-task and different-task masks based on the task indices. For samples from the same task, their similarity is encouraged to exceed a margin $\delta$ above the random baseline $(1/K)$, while for different tasks their similarity should be lower than a margin below $(1/K)$.

$$\mathcal{L}_{same} = \frac{1}{|M_{\text{same}}|} \sum_{(i,j) \in M_{\text{same}}} [max(0, (\frac{1}{K} + \delta) - S_{ij})],$$

$$\mathcal{L}_{diff} = \frac{1}{|M_{\text{diff}}|} \sum_{(i,j) \in M_{\text{diff}}} [max(0, S_{ij} - (\delta - \frac{1}{K}))],$$
(5)

where $M_{\text{same}}$ and $M_{\text{diff}}$ denote the index sets of sample pairs from the same and different tasks, respectively. The overall loss is then given by $\mathcal{L}_{cluster} = \mathcal{L}_{same} + \mathcal{L}_{diff}$. This router clustering enforces M-MoE router to produce consistent expert selections within the same modality while maintaining discriminative routing across modalities, thereby promoting both intra-modality coherence and inter-modality diversity. As a result, the model learns more discriminative and robust multimodal representations.

### 3.4. Discussion

**Why is a single MoE insufficient?** Although many previous works (Tan et al., 2025; Cai et al., 2025) have attempted to use a single MoE to learn more diverse features, motion information and modality features are two heterogeneous types of information. Using a single MoE to learn both simultaneously can lead to feature entanglement, where motion cues and domain-specific patterns may compete for the same expert neurons. As shown in Table 4, D-MoE outperforms a single MoE by a significant margin, confirming that decoupled learning is essential.

**How do T-MoE and M-MoE learn specialized features from identical inputs?** Guiding T-MoE and M-MoE to capture the corresponding features remains challenging. We propose the Expert Decoupling mechanism, which enforces T-MoE and M-MoE to learn distinct representations. By penalizing the cosine similarity between the outputs of the two branches, we force them into orthogonal functional subspaces. Since the tracking task inherently demands temporal consistency, T-MoE naturally gravitates toward motion-related features when its "overlap" with M-MoE is restricted. Since tracking inherently requires model to focus on motion information, T-MoE can naturally attend to motion-related features. As shown in Figure 5 and 4, T-MoE and M-MoE indeed learn different features, and T-MoE's expert selection patterns correlate with motion intensity. This demonstrates that T-MoE effectively captures motion cues.

**Why should M-MoE focus on specific modalities?** Different auxiliary modalities possess distinct statistical distributions. Router Cluster encourages each experts to capture

*Table 1.* **Details of *OneTrackerV2* variants**. We present four variants of *OneTrackerV2* with different input resolutions and model sizes. Meta Merger and DMoE introduce only marginal increases in parameters and computation. For clarity, the parameters and FLOPs of the text encoder are omitted here.

| Model | Search Resolution | Template Resolution | # Params (M) | FLOPs (G) | Speed (FPS) |
|---|---|---|---|---|---|
| *OneTrackerV2*-B224 | $224 \times 224$ | $112 \times 112$ | 80.2 | 23.8 | 72.4 |
| *OneTrackerV2*-B384 | $384 \times 384$ | $192 \times 192$ | 80.2 | 70.0 | 42.2 |
| *OneTrackerV2*-L224 | $224 \times 224$ | $112 \times 112$ | 271.1 | 77.7 | 46.6 |
| *OneTrackerV2*-L384 | $384 \times 384$ | $192 \times 192$ | 271.1 | 227.9 | 23.4 |

high-level modality-shared and modality-specific abstractions. As shown in Figure 4, M-MoE develops a hierarchical preference: certain experts respond to RGB features across all tasks, while others specialize in modality signatures. Without Router Cluster, the M-MoE would collapse into a generic feature extractor, failing to exploit the unique advantages of auxiliary sensors.

**Differences from Prior Works** Recent unified trackers (e.g., SUTrack (Chen et al., 2025)) utilizes naive token concatenation to handle multiple modalities, it often suffers from feature entanglement and performance collapse when modalities are missing. Our Meta Merger ensure modality robustness. Moreover, instead of processing all information in a single entangled stream, our architecture explicitly decouples tracking-specific motion cues from sensor-specific modality features. Compared with MoE-based models such as (Tan et al., 2025; Cai et al., 2025) that use experts primarily to increase model capacity or for domain adaptation, our DMoE is a structural solution to the heterogeneous objective conflict. By utilizing explicit decoupling and clustering losses, we ensure that the increased capacity is functionally partitioned between temporal reasoning and multimodal fusion, rather than just adding redundant parameters.

### 3.5. Optimization of OneTrackerV2

**Loss Function.** Following (Ye et al., 2022; Chen et al., 2025), we adopt a similar loss formulation to supervise *OneTrackerV2*, and the overall objective is defined as:

$$\mathcal{L} = \mathcal{L}_{class} + \lambda_G \mathcal{L}_{IoU} + \lambda_{L_1} \mathcal{L}_{L_1} + \mathcal{L}_{task} + \lambda_{dis} \mathcal{L}_{dis} \\ + \lambda_{cluster} \mathcal{L}_{cluster} + \lambda_{balance} \mathcal{L}_{balance}, \tag{6}$$

where $\mathcal{L}_{balance}$ is the balance loss, $\mathcal{L}_{class}$ and $\mathcal{L}_{task}$ is the same as that in (Chen et al., 2025), and $\lambda_G$, $\lambda_{L_1}$, $\lambda_{dis}$, $\lambda_{cluster}$, and $\lambda_{balance}$ are the hyperparameter with default values $\lambda_G = 2, \lambda_{L_1} = 5, \lambda_{dis} = 0.1, \lambda_{cluster} = 1$.

**Stochastic Modality Perturbation.** To enhance robustness and mitigate over-reliance on specific modalities, we introduce two stochastic training strategies. First, we apply modality replacement, where RGB and multimodal inputs are randomly swapped, encouraging model to learn modality-invariant representations. Second, inspired

by (Tan et al., 2025), we also perform random modality masking, where either the RGB or multimodal input is randomly masked. Together, these strategies expose Meta Merger to diverse modality configurations, enabling it to learn more discriminative and robust embeddings, thereby improving overall model robustness.

## 4. Experiments

### 4.1. Implementation Details

**Model Structure.** We develop a series of models trained with different model sizes and input resolutions to provide four variants of *OneTrackerV2*. All variants adopt HiViT (Zhang et al., 2022) initialized with Fast-iTPN (Tian et al., 2024) as the encoder. Number of experts $k$ and rank $r$ are set as 2 and 16. Detailed statistics of each model, including the number of parameters, FLOPs, and inference speed, are presented in Table 1. In addition, we employ CLIP-L (Radford et al., 2021) as the text encoder to extract language features. For tasks without text input, the parameters of the text encoder can be omitted.

**Training and Inference.** *OneTrackerV2* is trained on the combination of RGB and RGB+X tracking datasets, including LaSOT (Fan et al., 2019), TrackingNet (Muller et al., 2018), GOT-10k (Huang et al., 2019), COCO (Lin et al., 2014), VASTTrack (Peng et al., 2024), DepthTrack (Yan et al., 2021b), VisEvent (Wang et al., 2023), LasHeR (Li et al., 2021), and TNL2K (Wang et al., 2021). *OneTrackerV2* is optimized using AdamW (Loshchilov & Hutter, 2017) with a total of 300 training epochs, each consisting of 100,000 sampled image pairs. During training, we sample and mix data across these datasets. During inference, we apply a Hanning window penalty, following previous works (Ye et al., 2022). The template is updated every 25 frames, conditioned on a confidence threshold of 0.7.

### 4.2. Comparisons with the State-of-the-Art

**Main Results.** We evaluate *OneTrackerV2* against state-of-the-art RGB and RGB+X trackers across 5 tasks 12 benchmarks in Table 2. OneTrackerV2 consistently outperforms existing methods in terms of accuracy across all datasets. Specially, on RGB tracking benchmarks, *OneTrackerV2* surpasses prior RGB-only models, demonstrating that the unified design does not compromise single-modality performance. for RGB+X tracking, *OneTrackerV2* requires only a single training process and a unified parameters, yet still outperforms existing approaches that rely on task-specific architectures or downstream fine-tuning. Notably, on TNL2K, *OneTrackerV2* achieves 69.5 AUC. By unifying training, architecture, and parameters across diverse RGB and RGB+X tasks, *OneTrackerV2* not only achieves state-of-the-art accuracy but also delivers strong robustness and scalability.

*Table 2.* **Comparison with state-of-the-art trackers.** *OneTrackerV2* outperforms existing methods across 5 tasks and 12 benchmarks with a single training process, unified architecture, and shared parameters. We also compare whether each method supports multimodal inputs (MultiModal), employs a unified parameter (Unified Param), and is trained with a single step (Onestep Training).

| | | RGB Track | | | | | | | | | | | | |
|---|---|---|---|---|---|---|---|---|---|---|---|---|---|---|
| Method | Multi Modal | LaSOT | | | LaSOT$_{ext}$ | | | TrackingNet | | | GOT-10k | | | UAV123 | NFS |
| | | AUC | P$_{Norm}$ | P | AUC | P$_{Norm}$ | P | AUC | P$_{Norm}$ | P | AO | SR$_{0.5}$ | SR$_{0.75}$ | AUC | AUC |
| *OneTrackerV2*-B224 | ✓ | 74.1 | 83.6 | 81.0 | 53.2 | 64.3 | 60.8 | 86.3 | 90.6 | 86.2 | 78.4 | 87.8 | 79.6 | 70.8 | 70.5 |
| *OneTrackerV2*-B384 | ✓ | 75.4 | 85.4 | 83.8 | 54.1 | 65.1 | 61.3 | 87.2 | 91.1 | 87.6 | 79.6 | 89.4 | 81.3 | 71.1 | 70.9 |
| *OneTrackerV2*-L224 | ✓ | 74.9 | 85.0 | 83.1 | 54.8 | 65.8 | 62.2 | 87.5 | 91.6 | 88.3 | 80.5 | 90.2 | 82.7 | 70.8 | 70.6 |
| *OneTrackerV2*-L384 | ✓ | 76.1 | 86.0 | 84.4 | 55.2 | 66.1 | 62.9 | 88.6 | 92.5 | 89.0 | 81.3 | 91.8 | 83.9 | 71.0 | 70.8 |
| SUTrack-B224 (Chen et al., 2025) | ✓ | 73.2 | 83.4 | 80.5 | 53.1 | 64.2 | 60.5 | 85.7 | 90.3 | 85.1 | 77.9 | 87.5 | 78.5 | 70.9 | 69.8 |
| SUTrack-B384 (Chen et al., 2025) | ✓ | 74.4 | 83.9 | 81.9 | 52.9 | 63.6 | 60.1 | 86.5 | 90.7 | 86.8 | 79.3 | 88.0 | 80.0 | 70.4 | 69.3 |
| SUTrack-L224 (Chen et al., 2025) | ✓ | 73.5 | 83.3 | 80.9 | 54.0 | 65.3 | 61.7 | 86.5 | 90.9 | 86.7 | 81.0 | 90.4 | 82.4 | 70.9 | 69.8 |
| SUTrack-L384 (Chen et al., 2025) | ✓ | 75.2 | 84.9 | 83.2 | 53.6 | 64.2 | 60.5 | 87.7 | 91.7 | 88.7 | 81.5 | 89.5 | 83.3 | 70.4 | 69.3 |
| ARPTrack-B256 (Liang et al., 2025) | ✗ | 72.6 | 81.4 | 78.5 | 52.0 | 62.9 | 58.7 | 85.5 | 90.0 | 85.3 | 77.7 | 87.3 | 74.3 | 71.7 | 67.4 |
| SPMTrack-B384 (Cai et al., 2025) | ✗ | 74.9 | 84.0 | 81.7 | - | - | - | 86.1 | 90.2 | 85.6 | 76.5 | 85.9 | 76.3 | 71.7 | 67.4 |
| ODTrack-B384 (Zheng et al., 2024) | ✗ | 73.2 | 83.2 | 80.6 | 52.4 | 63.9 | 60.1 | 85.1 | 90.1 | 84.9 | 77.0 | 87.9 | 75.1 | - | - |
| LoRAT-B378 (Lin et al., 2024) | ✗ | 72.9 | 81.9 | 79.1 | 53.1 | 64.8 | 60.6 | 84.2 | 88.4 | 83.0 | 73.7 | 82.6 | 72.9 | 71.9 | 66.6 |
| ARTrackV2-256 (Bai et al., 2024) | ✗ | 71.6 | 80.2 | 77.2 | 50.8 | 61.9 | 57.7 | 84.9 | 89.3 | 84.5 | 75.9 | 85.4 | 72.7 | 69.9 | 67.6 |
| OneTracker-384 (Hong et al., 2024b) | ✓ | 70.5 | 79.9 | 76.5 | - | - | - | 83.7 | 88.4 | 82.7 | - | - | - | - | - |
| ARPTrack-L384 (Liang et al., 2025) | ✗ | 74.2 | 83.4 | 81.7 | 54.2 | 64.4 | 61.2 | 86.6 | 91.1 | 87.4 | 81.5 | 90.6 | 80.5 | 71.7 | 67.4 |
| SPMTrack-L384 (Cai et al., 2025) | ✗ | 76.8 | 85.9 | 84.0 | - | - | - | 86.9 | 91.0 | 87.2 | 80.0 | 89.4 | 79.9 | - | - |
| LoRAT-L378 (Lin et al., 2024) | ✗ | 75.1 | 84.1 | 82.0 | 56.6 | 69.0 | 65.1 | 85.6 | 89.7 | 85.4 | 77.5 | 86.2 | 78.1 | 72.5 | 66.7 |
| ODTrack-L384 (Zheng et al., 2024) | ✗ | 74.0 | 84.2 | 82.3 | 53.9 | 65.4 | 61.7 | 86.1 | 91.0 | 86.7 | 78.2 | 87.2 | 77.3 | - | - |
| ARTrackV2-L384 (Bai et al., 2024) | ✗ | 73.6 | 82.8 | 81.1 | 53.4 | 63.7 | 60.2 | 86.1 | 90.4 | 86.2 | 79.5 | 87.8 | 79.6 | 71.7 | 68.4 |

| | | | RGB+D Track | | | RGB+T Track | | | | RGB+E Track | | RGB+N Track | | | |
|---|---|---|---|---|---|---|---|---|---|---|---|---|---|---|---|
| Method | Unified Param | Onestep Training | DepthTrack | | | LasHeR | | RGBT234 | | VisEvent | | TNL2K | | OTB99 | |
| | | | F-Score | Re | Pr | AUC | P | MSR | MPR | AUC | P | AUC | P | AUC | P |
| *OneTrackerV2*-B224 | ✓ | ✓ | 65.9 | 66.7 | 65.8 | 60.6 | 75.5 | 69.5 | 91.3 | 63.0 | 79.8 | 65.8 | 70.2 | 72.5 | 94.8 |
| *OneTrackerV2*-B384 | ✓ | ✓ | 66.8 | 67.2 | 66.9 | 61.3 | 75.7 | 70.2 | 91.6 | 63.9 | 80.5 | 66.4 | 71.9 | 72.8 | 95.0 |
| *OneTrackerV2*-L224 | ✓ | ✓ | 66.6 | 67.7 | 66.4 | 62.5 | 77.7 | 72.0 | 94.5 | 64.7 | 81.9 | 67.5 | 73.0 | 72.9 | 94.6 |
| *OneTrackerV2*-L384 | ✓ | ✓ | 67.5 | 68.4 | 67.2 | 63.7 | 79.4 | 72.5 | 94.5 | 65.9 | 83.5 | 69.5 | 76.0 | 73.2 | 95.3 |
| SUTrack-B224 (Chen et al., 2025) | ✓ | ✓ | 65.1 | 65.7 | 64.5 | 59.9 | 74.5 | 69.5 | 92.2 | 62.7 | 79.9 | 65.0 | 67.9 | 70.8 | 93.4 |
| SUTrack-B384 (Chen et al., 2025) | ✓ | ✓ | 64.4 | 64.2 | 64.6 | 60.9 | 75.8 | 69.2 | 92.1 | 63.4 | 79.8 | 65.6 | 69.3 | 69.7 | 91.2 |
| SUTrack-L224 (Chen et al., 2025) | ✓ | ✓ | 64.3 | 64.6 | 64.6 | 61.9 | 77.0 | 70.8 | 94.6 | 64.0 | 80.9 | 66.7 | 70.3 | 72.7 | 94.4 |
| SUTrack-L384 (Chen et al., 2025) | ✓ | ✓ | 66.4 | 66.4 | 66.5 | 61.9 | 76.9 | 70.3 | 93.7 | 63.8 | 80.5 | 67.9 | 72.1 | 71.2 | 93.1 |
| CSTrack-L256 (Feng et al., 2025) | ✓ | ✗ | 65.8 | 66.4 | 65.2 | 60.8 | 75.6 | 70.9 | 94.0 | 65.2 | 82.4 | - | - | - | - |
| STTrack-B256 (Hu et al., 2025) | ✗ | ✗ | 63.3 | 63.4 | 63.2 | 60.3 | 76.0 | 66.7 | 89.8 | 61.9 | 78.6 | - | - | - | - |
| SeqTrackv2-L384 (Chen et al., 2023a) | ✓ | ✗ | 62.3 | 62.2 | 62.5 | 61.0 | 76.7 | 68.0 | 91.3 | 63.4 | 80.0 | 62.4 | 66.1 | 71.4 | 93.6 |
| SDSTrack-B256 (Hou et al., 2024) | ✗ | ✗ | 61.4 | 60.9 | 61.9 | 53.1 | 66.5 | 62.5 | 84.8 | 59.7 | 76.7 | - | - | - | - |
| UnTrack-B256 (Wu et al., 2024) | ✓ | ✗ | 61.0 | 60.8 | 61.1 | 51.3 | 64.6 | 62.5 | 84.2 | 58.9 | 75.5 | - | - | - | - |
| OneTracker (Hong et al., 2024b) | ✗ | ✗ | 60.9 | 60.4 | 60.7 | 53.8 | 67.2 | 64.2 | 85.7 | 60.8 | 76.7 | 58.0 | 59.1 | 69.7 | 91.5 |
| ViPT (Zhu et al., 2023) | ✗ | ✗ | 59.4 | 59.6 | 59.2 | 52.5 | 65.1 | 61.7 | 83.5 | 59.2 | 75.8 | - | - | - | - |

**Robustness Against Missing Modality.** Following Flex-Track (Bai et al., 2024), we conduct experiments on multiple RGB+X benchmarks under various missing-modality settings. As shown in Table 3, **OneTrackerV2** achieves significant improvements over previous methods. These results highlight not only the robustness of **OneTrackerV2** to modality absence, but also the effectiveness of Meta Merger in enabling stable and reliable multimodal fusion.

### 4.3. Ablation Study

We conduct ablation studies on HiViT-B (Zhang et al., 2022) to examine how the modules we propose contribute to building a unified multimodal tracker. Note that in all ablation experiments, the models are trained for only 180 epochs.

**Meta Merger.** To demonstrate the effectiveness of the proposed Meta Merger module, we conduct ablation experiments, and the results are shown in row #2 and #3 of Table 4. Compared with the baseline, integrating the Meta Merger yields more effective fusion of multi-modal features, leading to consistent performance gains. Besides, the computational

cost of Meta Merger is negligible (only additional 0.4% FLOPs and 0.9% parameters).

Moreover, under modality-missing scenarios, the combination of Meta Merger and Stochastic Perturbation significantly enhances the robustness of the model. This indicates that our design not only improves the overall representation ability in multi-modal fusion, but also equips model with robust adaptability to incomplete or noisy modality inputs.

We further validate the generalization potential of the Meta Merger by testing on an **unseen** modality excluded from the training phase. Using the CMOTB dataset (Liu et al., 2024), which consists of Near-Infrared (NIR) signals, we observed that **OneTrackerV2** achieves an impressive SR of 65.1 (see Table 5). Notably, this performance not only outperforms current state-of-the-art methods but also exceeds models explicitly trained on the CMOTB dataset (ProtoTrack and MAFNet). This outcome strongly underscores the exceptional generalization capability of the Meta Merger.

**Dual Mix-of-Experts.** We further investigate the impact of the proposed Dual MoE design, with results summarized in

*Table 3.* **Comparison on missing-modality benchmarks.** We evaluate *OneTrackerV2* and existing RGB+X trackers under scenarios where certain modalities are absent. Thanks to the proposed Meta Merger, *OneTrackerV2* consistently outperforms prior methods, demonstrating superior robustness to modality missing in multimodal tracking.

| Method | Unified Param | Onestep Training | DepthTrack$_{miss}$ | | | LasHeR$_{miss}$ | | RGBT234$_{miss}$ | | VisEvent$_{miss}$ | |
|---|---|---|---|---|---|---|---|---|---|---|---|
| | | | F-score | Re | Pr | AUC | P | MSR | MPR | AUC | P |
| ***OneTrackerV2*-B224** | ✓ | ✓ | 56.9 | 55.4 | 58.6 | 52.7 | 66.2 | 63.5 | 85.5 | 54.3 | 71.5 |
| STTrack (Hu et al., 2025) | ✗ | ✗ | 49.9 | 48.8 | 51.0 | 44.9 | 54.5 | 54.2 | 73.8 | 49.7 | 65.5 |
| SUTrack (Chen et al., 2025) | ✓ | ✓ | 49.5 | 47.3 | 51.9 | 47.6 | 58.3 | 60.8 | 82.0 | 50.5 | 66.6 |
| SeqTrackv2 (Chen et al., 2023a) | ✗ | ✗ | 45.0 | 40.9 | 50.0 | 39.9 | 50.0 | 49.9 | 70.8 | 43.1 | 57.6 |
| SDSTrack (Hou et al., 2024) | ✗ | ✗ | 46.7 | 42.0 | 52.7 | 43.1 | 52.5 | 48.8 | 67.0 | 46.9 | 62.6 |
| ViPT (Zhu et al., 2023) | ✗ | ✗ | 44.4 | 40.5 | 46.6 | 34.0 | 40.1 | 39.4 | 52.4 | 43.2 | 57.2 |
| MCITrack (Kang et al., 2025) | ✗ | ✗ | 49.7 | 42.9 | 59.1 | 40.0 | 34.2 | 40.9 | 53.6 | 36.5 | 49.9 |

*Table 4.* **Ablation study.** We investigate the impact of each module on *OneTrackerV2*.

| # | Method | FPS ↑ | Params (M) | FLOPs (G) | LaSOT | TNL2K | LasHeR | DepthTrack | VisEvent | Average | DepthTrack$_{miss}$ | LasHeR$_{miss}$ |
|---|---|---|---|---|---|---|---|---|---|---|---|---|
| 1 | Baseline | 95 | 70.0 | 23.0 | 69.2 | 61.3 | 50.4 | 51.7 | 53.5 | 57.2 | 48.4 | 43.7 |
| 2 | + Meta Merger | 90(−5.3%) | 70.6(+0.9%) | 23.1(+0.4%) | 71.3 | 62.5 | 55.7 | 62.1 | 58.8 | 62.1 | 51.8 | 47.9 |
| 3 | + Stochastic Perturbation | 90(−5.3%) | 70.6(+0.9%) | 23.1(+0.4%) | 71.0 | 62.4 | 55.6 | 62.3 | 58.5 | 62.0 | 53.2 | 48.6 |
| 4 | + Single Mixture-of-Experts | 83(−11.6%) | 75.4(+7.7%) | 23.5(+2.2%) | 71.4 | 62.9 | 56.2 | 62.9 | 58.4 | 62.4 | 53.6 | 49.3 |
| 5 | + Dual Mixture-of-Experts | 72(−24.2%) | 80.2(+14.6%) | 23.8(+3.5%) | 71.6 | 63.3 | 56.8 | 63.2 | 59.2 | 62.8 | 54.1 | 49.8 |
| 6 | + Expert Decoupling | 72(−24.2%) | 80.2(+14.6%) | 23.8(+3.5%) | 71.6 | 63.3 | 56.8 | 63.2 | 59.2 | 62.8 | 54.7 | 50.2 |
| 7 | + Router Cluster | 72(−24.2%) | 80.2(+14.6%) | 23.8(+3.5%) | 72.0 | 63.7 | 57.5 | 64.2 | 60.2 | 63.5 | 56.9 | 52.7 |

*Table 5.* **Results on CMOTB.** *OneTrackerV2* outperforms existing methods even on unseen modality.

| Method | PR | NPR | SR |
|---|---|---|---|
| ***OneTrackerV2*** | 65.7 | 74.8 | 65.1 |
| SUTrack (Chen et al., 2025) | 62.9 | 70.2 | 62.8 |
| ProtoTrack (Liu et al., 2025) | 60.5 | 69.7 | 59.7 |
| MAFNet (Liu et al., 2024) | 55.1 | 66.4 | 56.1 |
| SDSTrack (Hou et al., 2024) | 33.9 | 41.5 | 50.2 |
| GRM (Gao et al., 2023) | 41.1 | 47.7 | 43.4 |
| MixFormerV2 (Cui et al., 2023) | 41.5 | 48.5 | 44.2 |

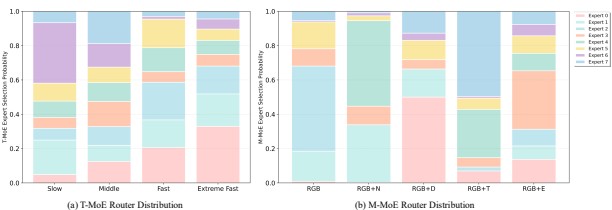

(a) T-MoE Router Distribution     (b) M-MoE Router Distribution

*Figure 4.* **Visualization of D-MoE Router Distribution.** We show the distribution of the T-MoE router under different object motion speeds in (a), and the distribution of the M-MoE router under different modalities in (b).

Table 4. When we simply increase the capacity of a single MoE (row #4), model exhibits only moderate performance improvement. Extending this design to a straightforward Dual MoE brings only marginal additional gains, indicating that naively stacking MoEs is insufficient to fully exploit their potential. Then, we incorporate two key enhancements: Expert Decoupling and Router Clustering. With these two modifications, Dual MoE achieves significant performance gains, demonstrating the effectiveness and necessity of our Dual MoE. Moreover, the introduction of Dual MoE only incurs an interpolation of 14.6% parameters and 3.5% FLOPs, which is a relatively small price to pay given the significant performance gains achieved.

Moreover, we present the visualization results of DMoE in Figure 5. The results show that the shared expert, T-MoE, and M-MoE each learn distinct types of features. The shared expert primarily captures general representations, whereas T-MoE focuses on motion information and M-MoE emphasizes modality-specific cues.

To further investigate and understand the behavior of D-MoE, we visualize the expert-selection distributions of T-MoE and M-MoE under different motion speeds and modal-

ities. Based on the magnitude of the object's center displacement between consecutive frames, we categorize object motion into four levels (slow, middle, fast, and extreme fast),—and visualize the routing distribution of T-MoE in Figure 4 (a). A clear correlation emerges between the routing pattern and motion speed. When the object moves slowly, T-MoE tends to select the seventh and eighth experts; when motion becomes faster, it increasingly prefers the first expert. Notably, as motion speed rises, the proportion of selections for the first expert grows significantly, indicating that T-MoE has learned to adaptively select experts based on motion characteristics.

Figure 4 (b) further shows that M-MoE exhibits modality-dependent routing preferences. For instance, in the RGB tracking task, M-MoE predominantly selects the third expert, while in the RGB+T task, it rarely chooses the third expert and instead strongly favors the eighth expert.

Overall, the results in Table 4, Figures 5 and 4 consistently demonstrate that D-MoE successfully learns two distinct and complementary types of features. The combination of expert decoupling and router clustering encourages a clear

*Table 6.* **Compression Performance.** We compress *OneTrackerV2-B224* into a 6-layer variant (*OneTrackerV2*-B224-Compress) following CompressTracker (Hong et al., 2024a). *OneTrackerV2*-B224-Compress achieves the balance between accuracy and efficiency.

| Method | LaSOT | | | DepthTrack | | | LasHeR | | VisEvent | | TNL2K | | FPS |
|---|---|---|---|---|---|---|---|---|---|---|---|---|---|
| | AUC | $P_{norm}$ | P | F | Re | Pr | AUC | P | AUC | P | AUC | P | |
| *OneTrackerV2*-B224-Compress | 73.0 | 83.1 | 81.1 | 64.6 | 65.0 | 65.2 | 59.7 | 74.9 | 62.0 | 79.2 | 64.7 | 68.5 | 159 |
| SUTrack-T224 (Chen et al., 2025) | 69.6 | 79.3 | 75.4 | 61.7 | 62.1 | 61.2 | 53.9 | 66.7 | 58.8 | 75.7 | 60.9 | 62.3 | 100 |
| STTrack-B256 (Hu et al., 2025) | - | - | - | 63.3 | 63.4 | 63.2 | 60.3 | 76.0 | 61.9 | 78.6 | - | - | 36 |
| OneTracker-384 (Hong et al., 2024b) | 70.5 | 79.9 | 76.5 | 60.9 | 60.4 | 60.7 | 53.8 | 67.2 | 60.8 | 76.7 | 58.0 | 59.1 | - |
| SeqTrackerV2-B224 (Chen et al., 2023b) | 69.9 | 79.7 | 76.3 | 63.2 | 63.4 | 62.9 | 55.8 | 70.4 | 61.2 | 78.2 | 57.5 | 59.7 | 40 |

*Figure 5.* **Visualization of D-MoE.** We present the visualization results of T-MoE and M-MoE under different RGB and RGB+X tracking tasks. It can be observed that the shared expert, T-MoE, and M-MoE have learned distinct features.

separation in the roles of the two MoE modules: T-MoE exhibits strong sensitivity to motion dynamics, while M-MoE emphasizes modality-related cues.

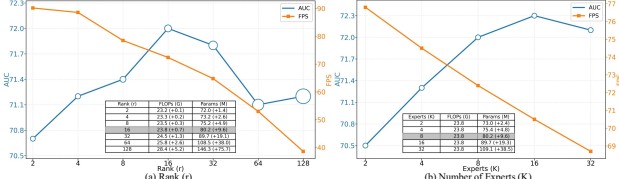

*Figure 6.* **Analysis of expert hyperparameters.** We show the impact of (a) different rank ($r$) and (b) different number of experts ($K$) on model parameters, computational cost, FPS, and accuracy.

**Hyper Parameters of Experts.** We investigate the impact of hyper-parameters in DMoE, specifically the expert projection rank ($r$) and the number of experts ($K$). As illustrated in Figure 6, increasing the rank consistently improves model performance in the beginning, which indicates that higher-rank projections enhance the expressive capacity of the Mixture-of-Experts and allow the model to capture richer patterns. However, when the rank exceeds 16, the performance starts to drop slightly. This phenomenon suggests that overly large ranks may introduce redundancy into the learned representations, which limits further improvement. Moreover, as the rank increases, inference speed inevitably decreases, clearly highlighting a trade-off between efficiency and accuracy. Figure 6 (a) demonstrates that setting the rank to 16 achieves the optimal balance between inference efficiency and tracking accuracy. Similarly, Figure 6 (b) examines the effect of the number of experts. Enlarging the expert count generally leads to better performance, since additional experts expand the representational space. Nevertheless, excessively increasing the number of

experts also results in prohibitively high parameter overhead and computational burden, which is impractical in real applications. Taking both accuracy and efficiency into account, we therefore adopt 8 experts as the optimal configuration for our *OneTrackerV2*, striking a desirable balance between representational capacity and model complexity.

**Compression Effectiveness.** Following CompressTracker (Hong et al., 2024a), we compress our *OneTracker*-B224 into a lightweight variant *OneTracker*-B224-Compress with only six layers. As shown in Table 6, the compressed model achieves a $2.2\times$ speedup while incurring only about a $2\%$ performance drop on both RGB and RGB+X tracking benchmarks. Specifically, *OneTrackerV2*-B224-Compress reaches 159 FPS, surpassing SuTrack-T224 (Chen et al., 2025) in both speed and accuracy. *OneTrackerV2*-B224-Compress outperforms SuTrack-T224 by 3.4 AUC on LaSOT (73.0 *vs* 69.6) and by 5.8 AUC on LasHeR (59.7 *vs* 53.9), which demonstrates the effectiveness of *OneTrackerV2* under compression.

## 5. Conclusion

In this work, we present *OneTrackerV2*, a unified paradigm for multimodal tracking. Instead of designing modality-specific structure or relying on multi-stage fine-tuning, we demonstrate that a single framework with Meta Merger for modality unification and DMoE for decoupled expert learning can achieve robust multimodal tracking with one-step training. Extensive experiments show that *OneTrackerV2* not only surpasses existing models in both RGB and RGB+X tracking tasks, but also remains robust under missing modalities and effective in compressed settings.

## Impact Statement

We propose *OneTrackerV2*, a unified framework for multimodal visual object tracking, which yields the following impacts:

**Advancement in Intelligent Systems.** By introducing the Meta Merger and Dual Mixture-of-Experts (DMoE), *OneTrackerV2* provides a scalable and efficient paradigm for integrating heterogeneous sensory data (e.g., RGB, Depth, Thermal, and Language). This advancement is crucial for developing robust autonomous systems, such as self-driving vehicles and search-and-rescue robots, which must operate reliably across diverse and adverse environmental conditions (e.g., night, fog, or sensor failure).

**Resource Efficiency and Sustainability.** Traditional multimodal tracking often requires training and deploying separate models for different sensor combinations, leading to significant computational overhead and carbon footprints. Our unified approach enables a single model to handle multiple modality tasks, substantially reducing the energy consumption associated with model training and the memory requirements for deployment on edge devices.

**Ethical Considerations and Privacy.** While enhanced tracking capabilities offer significant benefits for safety and automation, they also raise potential concerns regarding surveillance and privacy. The ability to track objects accurately across multiple modalities (e.g., using thermal imaging to track people in the dark) could be misused for unauthorized monitoring. We encourage the community to deploy such technologies within established legal and ethical frameworks, prioritizing privacy-preserving applications.

## Acknowledgements

This work was supported by National Natural Science Foundation of China (No.62576109), Scientific and Technological Innovation Action Plan of Shanghai Science and Technology Committee (No.25511104402).

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
