# OpenReview forum: "Unified Multimodal Visual Tracking with Dual Mixture-of-Experts"
_ICML.cc/2026/Conference — ICML 2026 regular_

### Official Review · Reviewer_aKxv · 2026-02-24

**Soundness:** 3
**Presentation:** 3
**Significance:** 3
**Originality:** 3
**Overall Recommendation:** 4
**Confidence:** 3

**Summary:**

the paper introduce onetrackerv2, a unified visual object tracking method with state-of the-art performance across five RGB and RGB+X tracking tasks and 12 benchmarks.

**Compliance With Llm Reviewing Policy:**

Affirmed.

**Final Justification:**

The rebuttal addressed my concerns. I will keep my positive rating.

**Key Questions For Authors:**

- Does the author attempted other text encoder such as BERT?

**Limitations:**

Yes

**Strengths And Weaknesses:**

strength
- paper is clearly written and easy to follow, the motivation behind unifying multimodal tracking into a single architecture is clearly explained. The inclusion of detailed figures further improves readability.
- clear motivation, the paper address the chlalenge of modality-specific models and multi-stage training pipelines in existing methods.
- Strong empirical performance on existing benchmarks (5 tasks, 12 datasets), including both RGB and RGB+X settings.

---

weakness
- Limited text encoder design and potential bottleneck. The paper adopts clip as text encoder, which might resulted in failure on more fine-grained understanding on long text description of target object. This may limit performance in RGB+N tracking scenarios where precise attribute grounding is required.
- no comparison with vision foundation model such as as SAM2 [1] or its extensions (e.g., SAMURAI [2], SAM2Long [3]). SAM3 [4] also has the language tracking ability, which should be included as a baseline.
- Insufficient exploration of text encoder alternatives : does the author attempted other text encoder such as BERT?

---

[1] Ravi, Nikhila, et al. "Sam 2: Segment anything in images and videos."

[2] Yang, Cheng-Yen, et al. "Samurai: Adapting segment anything model for zero-shot visual tracking with motion-aware memory."

[3] Ding, Shuangrui, et al. "Sam2long: Enhancing sam 2 for long video segmentation with a training-free memory tree." Proceedings of the IEEE/CVF International Conference on Computer Vision. 2025.

[4] Carion, Nicolas, et al. "Sam 3: Segment anything with concepts."

---

> ### Author Rebuttal · Authors · 2026-03-27
>
> Thank you for your constructive suggestion. We hope the following response resolves your concern.
>
>
> > **Q1: Text Encoder**
> We chose the CLIP text encoder over others like BERT based on several key considerations tailored to the unified tracking task:
>
> 1. **Vision-Language Alignment:** Unlike BERT, which is trained solely on text, CLIP is pre-trained on large-scale image-text pairs. This provides a "vision-aware" textual embedding space that naturally aligns with the visual tokens in our tracker. This pre-aligned feature space significantly eases the burden on the Meta Merger when mapping linguistic descriptions and visual features into a unified space.
>
> 2. **Tracking-Specific Requirements:** Most natural language tracking (RGB+N) benchmarks focus on identifying target objects through physical attributes (e.g., "a person in a red shirt"). CLIP's contrastive learning objective is specifically optimized for this kind of attribute-level grounding, whereas BERT's strengths in deep linguistic logic are less critical for current tracking scenarios.
>
> 3. **Architectural Flexibility:** It is important to note that our Meta Merger and Dual MoE framework are encoder-agnostic. The architecture is flexible enough to incorporate more complex encoders (like BERT or even LLMs) if future benchmarks demand more sophisticated linguistic reasoning.
>
>
> 4. **Comparative Experiments:** We conduct experiments by replacing the CLIP text encoder with BERT while keeping other components of OneTracker V2 identical. As shown in the table below, using BERT as the text encoder leads to a performance drop compared to CLIP:
>
> Text Encoder | AUC (TNL2K)
> ---- | ----
> CLIP (OneTrackerV2) | 63.0
> BERT | 61.8
>
> We will include these experimental results and a detailed discussion in the revised version.
>
>
> > **Q2: Comparison with Vision Foundation Models like SAM 2/3**
>
> We add comparisons with the SAM2 and SAM3 models in the table below. It can be seen that compared with SAM2 and SAM3, our OneTrackerV2 achieves higher accuracy, which demonstrates the superiority of OneTrackerV2. We will add performance comparisons with the SAM2/3 models in the revised version later.
>
>
> Model | LaSOT | GOT-10k
> ---- | ---- | ----
> OneTrackerV2 | 76.1  | 81.3
> SAM2-L | 70.0 | 80.7
> SAM2Long | 73.9 | 81.1
> SAMURAI-L | 74.2   | 81.7
>
>
> Model | TNL2K
> ---- | ----
> OneTrackerV2 | 69.5
> SAM3-L | 64.2

---

> > ### Author Rebuttal · Reviewer_aKxv · 2026-04-01
> >
> > Thanks the author for the rebuttal especially the additional experiments with BERT. I will keep my positive rating.

---

### Official Review · Reviewer_G329 · 2026-02-24

**Soundness:** 4
**Presentation:** 4
**Significance:** 3
**Originality:** 3
**Overall Recommendation:** 5
**Confidence:** 3

**Summary:**

This paper tackles the problem of multi-modal visual object tracking. Such tasks use a variety of multi-modal input such as RGB, depth, thermal, event, and language features. Previous work often trains different models on each modality, however such approaches often lead to reduced discriminability. To solve these problems the paper introduces "Meta Merger", which embeds all modalities into a shared embedding. The embeddings are then fed into a vision transformer before applying a Dual Mixture of Experts to decouple the two groups of modalities. The authors opt for two experts, one that focuses on spatio-temporal patterns and another that focuses on multi-modal knowledge fusion. The final model with a tracking head is coined the "One-TrackerV2" and is tested on 12 benchmarks, where it achieves superior or comparable results to approaches introduced in recent literature.

**Compliance With Llm Reviewing Policy:**

Affirmed.

**Final Justification:**

My concerns were fully addressed, so I'm keeping my positive score of the submission.

**Key Questions For Authors:**

1. Why is the DMoE architecture the right approach, given that SUTrack achieves similar performance by only levering a shared embeding.
2. Could the M-MoE be split into more then one experts, then enforce a higher dimensional orthogonality constraint in the MoE loss?
3. How small of an embedding could you generate before performance begins to drastically decrease?
4. The DMoE introduces orhogonality-based decoupling, are these two components independent enough to justify strong orthogonality? Or should the DMoE loss reward partial entanglement?

**Limitations:**

yes

**Strengths And Weaknesses:**

- This work seems very similar to SUTrack, which also creates a unified input representation using multiple modalities. It seems as if the primary difference is the introduction of a MoE step, however the experimental results (specifically the ablation study) don't seem to support that the MoE step adds a whole lot of value. I would suggest the additional experiments to support the use of MoE beyond the qualitative discussion in section 3.4.
- The authors outline 5 limitations that past work has suffered from, however it's not clear whether such limitations truly create downstream issues with respect to the final performance of such models (i.e It's not clear whether these issues actually result in "reduced discriminability")
- I am also curious as to how the size of the Meta-Merger embeddings impact performance. If your claims are true, then this approach seems to offer an interesting opportunity to compress multi-modal data into small and informative embeddings. How small of an embedding could you generate before performance begins to drastically decrease?
- Its not clear why two experts is the best choice. It seems as if T-MoE has a clearly defined task, wheras the M-MoE aims to explain all other "left over" multi-modal information. Could the M-MoE be split into more then one MoE?

---

> ### Author Rebuttal · Authors · 2026-03-27
>
> Thanks for your insightful comments and the recognition of our OneTrackerV2's performance.
>
> > **Q1: DMoE**
>
> In fact, OneTrackerV2 differs significantly from SUTrack, and we have verified the effectiveness and underlying mechanism of DMoE and Meta Merger through extensive experiments and analyses.
>
> 1. **Meta Merger.**
> SUTrack directly concatenates multi-modal features without a shared embedding, whereas we employ Meta Merger to map them into a unified space, improving robustness to missing modalities. As shown in ***Table 3***, OneTrackerV2 outperforms SUTrack by **7.4 **on DepthTrack miss, demonstrating the effectiveness of our approach.
>
> 2. **DMoE**
> SUTrack learns spatiotemporal and modality-specific features in a single space, which may cause conflicts. In contrast, we address this issue with DMoE, whose effectiveness is validated through extensive experiments.
>
> - **Accuracy.** As shown in ***Table 4***, DMoE brings an average performance gain of 1.4 without noticeable inference slowdown, demonstrating its effectiveness.
>
> - **Visualization**
> We present the feature visualization of DMoE under different tasks in ***Figure 5***. T-MoE and M-MoE have learned distinct features: T-MoE focuses on motion, while M-MoE concentrates on modality-specific characteristics, which validates our motivation.
>
>
> - **D-MoE Router Distribution** In addition, we visualize the router distribution of DMoE in ***Figure 6***. T-MoE’s router varies with motion speed, while M-MoE exhibits modality-specific expert preferences. This confirms that DMoE captures distinct motion and modality characteristics, aligning with our design motivation.
>
>
> In summary, OneTrackerV2 differs significantly from SUTrack. Our Meta Merger greatly improves the robustness in noisy scenarios, while DMoE decouples motion information and modality features, substantially enhancing tracking accuracy. We verify the effectiveness of Meta Merger and DMoE through **extensive experiments, visualization results, and analyses**, and their performance is highly consistent with our motivation. We will further summarize and emphasize the novelty of OneTrackerV2 and its differences from previous work in the revised version.
>
>
> > **Q2 "5 Limitations" and Discriminability**
>
> These five limitations do not directly reduce discriminability but significantly constrain model applicability:
>
> 1. **Multi-step training** can cause optimization drift, leading to suboptimal performance.
>
> 2. **Lack of a unified architecture** requires separate models for different modalities, reducing usability.
>
> 3. **Un-unified Parameters** weaken generalization due to modality-specific training.
>
>
> 4. **Vulnerability to missing modalities** does not affect clean scenarios but causes significant accuracy drops under noisy conditions. As shown in ***Table 3***, existing models degrade severely, while OneTrackerV2 remains robust.
>
> 5. **Feature Conflict** arises from jointly learning motion and modality information in a shared space. OneTrackerV2 effectively mitigates this issue, achieving an average performance gain of 6.3 across five tasks.
>
> Overall, these issues hinder generalization and scalability. In contrast, OneTrackerV2 addresses them with a unified training framework and achieves superior performance across tasks.
>
>
> > **Q3: Embedding Size**
>
> The embedding size is now aligned with the Transformer's hidden dimension. Thank you for your insightful suggestion. We will investigate reducing the embedding dimension to achieve multi-modal information compression in future work, so as to further decrease the number of parameters.
>
>
>
> > **Q4: Two MoE**
>
> We adopt a dual-structure to capture the fundamental split between spatio-temporal (“where”) and multimodal (“what”) information, using two MoE modules as sufficient representations. We think that two MoE modules are sufficient for representation.
>
> M-MoE focuses on mining shared and modality-specific features rather than modeling all multimodal information. As shown in ***Figure 5(b)***, expert selection reveals clear modality preferences (e.g., RGB-related tasks favor certain experts, while others capture modality-specific cues), demonstrating its ability to learn discriminative features.
>
> Although M-MoE could be extended to more modality-specific MoEs, this would increase model complexity, parameter size, and make orthogonality constraints harder to enforce, potentially leading to over-decoupling and reduced complementarity.
>
>
> > **Independence vs. Entanglement of DMoE**
>
> We believe the T-MoE and M-MoE should be mostly independent but weakly coupled. Our decoupling loss encourages orthogonality but does not strictly prevent the model from sharing low-level geometric features. Rewarding "partial entanglement" is an interesting direction that we plan to explore by softening the similarity penalty in future work.

---

> > ### Author Rebuttal · Reviewer_G329 · 2026-04-02
> >
> > My concerns have been fully addressed. I will keep my positive rating.

---

### Official Review · Reviewer_K79F · 2026-03-08

**Soundness:** 3
**Presentation:** 4
**Significance:** 3
**Originality:** 4
**Overall Recommendation:** 4
**Confidence:** 5

**Summary:**

This paper proposes OneTrackerV2, a unified multimodal visual object tracking framework, which targets key limitations of existing RGB and RGB+X tracking methods: low efficiency of modality-specific models, cumbersome multi-stage fine-tuning pipelines, and poor robustness against missing modalities. The core innovations lie in the Meta Merger module and the Dual Mixture of Experts (DMoE). The former enables efficient fusion of heterogeneous modal features into a shared embedding space, while the latter alleviates cross-modal feature conflicts by decoupling spatial-temporal relationship modeling and multimodal knowledge integration. Built upon a shared architecture and unified parameters, the framework is trained end-to-end in a single stage. It achieves state-of-the-art performance across 5 tracking tasks and 12 benchmark datasets, while maintaining strong robustness and inference efficiency under missing modalities and model compression scenarios. Extensive ablation studies and visualizations fully validate the effectiveness of each component.

**Compliance With Llm Reviewing Policy:**

Affirmed.

**Final Justification:**

My concerns are addressed, thus I keep the positive recommendation.

**Key Questions For Authors:**

1.	In the random modality perturbation strategy, what is the probability settings for modality replacement and modality masking.
2.	The paper contains minor issues of repetitive and redundant statement. It is recommended to refine and proofread the main text to improve readability.

**Limitations:**

In multimodal tracking, there exists the problem of modality data asynchrony (e.g., the capture time difference between RGB and depth frames), but the paper does not mention the model’s ability to handle this issue.

**Strengths And Weaknesses:**

Strengths
1.	OneTrackerV2 construct a scalable and robust multimodal tracking solution. Its one-stage training and parameter-sharing characteristics are highly valuable for practical deployment. Moreover, it even outperforms baseline models optimized specifically for RGB on RGB tracking tasks, demonstrating that the unified design does not sacrifice single-modal performance.
2.	Meta Merger achieves effective alignment and fusion of multimodal features with extremely low parameter and computational overhead, addressing the feature entanglement issue in simple concatenation and the efficiency problem in multi-branch designs. DMoE innovatively introduces expert decoupling loss and router clustering loss to realize explicit separation of spatial-temporal features and multimodal features, and its sparse activation design balances model capacity and inference speed.
3.	Experiments cover various scenarios including standard benchmarks, missing modalities, model compression, and generalization to unseen modalities. Ablation studies dissect the contribution of each module and loss function, and visualization analyses intuitively show the expert selection behavior of DMoE. The complete experimental design and reliable conclusions fully validate the effectiveness of the proposed method.

Weakness:
1.	OneTrackerV2 adopts mixed training on multiple datasets but does not specify the batch sampling strategy during training or the detailed settings of data augmentation. A brief supplement is suggested to improve the reproducibility of the experiments.
2.	The unseen modality generalization experiments are only validated on the Near-Infrared (NIR) modality, and the model’s generalization potential to other less common modalities is not discussed.
3.	For the expert decoupling loss in DMoE, only the squared loss based on cosine similarity is adopted, while other decoupling loss functions (such as mutual information loss and contrastive loss) are not explored.
4.	Some formulas in the paper lack detailed annotations (e.g., Lbalance in the loss function is only named without a brief explanation of its function). There are also minor flaws in the labeling of some figures and tables. A unified proofreading and revision is recommended.

---

> ### Author Rebuttal · Authors · 2026-03-27
>
> Thanks for your insightful comments and recognition of our work. We greatly appreciate your acknowledgment of the novelty of our work.
>
> > **Q1: Training Strategy and Data Augmentation**
>
> We sincerely apologize for the omission of these details in the initial submission.
>
> 1. **Sampling Strategy:** During mixed training, we adopt a balanced sampling strategy. Specifically, for each training batch, samples are drawn from the RGB, RGB+D, RGB+T, RGB+E, and RGB+N datasets with a ratio of [1:1:1:1:1], ensuring that the model assigns equal importance to each modality.
>
> 2. **Data Augmentation:** We employ standard data augmentation techniques, following the practices used in OSTrack and SUTrack.
>
> We appreciate your suggestion and will include a more detailed description of the training settings in the Appendix of the revised manuscript.
>
> > **Q2: Generalization to Unseen Modalities**
>
> We select the Near-Infrared (NIR) modality (CMOTB dataset) for validation because it represents a substantial domain shift from the visible and thermal spectra used during training. Achieving a success rate (SR) of 65.1, surpassing models specifically trained on NIR, demonstrates that our Meta Merger effectively maps diverse physical signals into a unified and robust embedding space, thereby exhibiting strong generalization capability.
>
> We validate the effectiveness of OneTrackerV2 across five multi-modal tracking tasks, including four in-domain tasks and one out-of-domain task. As these constitute the primary publicly available multi-modal tracking benchmarks, and no additional modal datasets are accessible, we select the NIR modality as the representative out-of-domain setting to evaluate generalization performance.
>
>
>
> > **Q3: Decoupling Loss Functions**
>
> We adopt the squared loss as the expert decoupling loss due to its simplicity, aiming to achieve effective expert decoupling with a straightforward formulation. The ablation results in Table 4 further validate the effectiveness of this design. We think that other types of loss functions could yield comparable performance, and we will supplement experiments with different loss functions in future work.
>
> > **Q4: Formulas, Annotations, and Writing**
>
> We thanks your meticulous check. We will add a formal definition and explicit annotations for all variables in the revised version. We will conduct a thorough professional proofreading of the manuscript to remove redundant statements and correct the labeling inconsistencies in the figures/tables mentioned.
>
>
>
> > **Q5: Modality Perturbation**
>
> In our random modality perturbation strategy, the probability for modality replacement (substituting one modality with another) is set to 0.3, and the probability for modality masking (simulating missing modalities) is set to 0.25.
>
>
> > **Q5: Modality Asynchrony**
>
> Currently, OneTrackerV2, like most state-of-the-art multimodal trackers, assumes that input frames are hardware-synchronized. However, we acknowledge that temporal asynchrony (time-lag between sensors) is a critical real-world challenge. We will add a dedicated paragraph to discuss this limitation

---

> > ### Author Rebuttal · Reviewer_K79F · 2026-04-02
> >
> > Thanks for authors' rebuttal. My concerns are addressed, thus I keep the positive recommendation.

---

### Decision · Program_Chairs · 2026-04-30

**Decision:**

Accept (regular)

**Comment:**

The submission received 3 reviews, 1x accept, 2x weak accept. The authors provided rebuttals. None of the reviewers recommends to reject. All reviewers agree that the strengths dominate the weaknesses.